# The Impact of Primary Tumor Location on Long-Term Oncological Outcomes in Patients with Upper Tract Urothelial Carcinoma Treated with Radical Nephroureterectomy: A Systematic Review and Meta-Analysis

**DOI:** 10.3390/jpm11121363

**Published:** 2021-12-14

**Authors:** Wojciech Krajewski, Łukasz Nowak, Bartosz Małkiewicz, Joanna Chorbińska, Paweł Kiełb, Adrian Poterek, Bartłomiej Sporniak, Michał Sut, Marco Moschini, Chiara Lonati, Roberto Carando, Jeremy Yuen-Chun Teoh, Keiichiro Mori, Krzysztof Kaliszewski, Tomasz Szydełko

**Affiliations:** 1University Center of Excellence in Urology, Department of Minimally Invasive and Robotic Urology, Wroclaw Medical University, 50-556 Wroclaw, Poland; wk@softstar.pl (W.K.); joanna.chorbinska@gmail.com (J.C.); pk.kielb@gmail.com (P.K.); adipoterek@gmail.com (A.P.); bart6b@gmail.com (B.S.); tomasz.szydelko1@gmail.com (T.S.); 2Department of Urology, Ministry of Interior and Administration Hospital in Gdansk, 80-104 Gdansk, Poland; doktor.sut@gmail.com; 3Department of Urology, Luzerner Kantonsspital, 6004 Lucerne, Switzerland; marco.moschini87@gmail.com (M.M.); roberto.carando@bluewin.ch (R.C.); 4Department of Urology, Spedali Civili of Brescia, 25123 Brescia, Italy; chiara.lonati@libero.it; 5Clinica Luganese Moncucco, 6900 Lugano, Switzerland; 6Clinica S. Anna, Swiss Medical Group, 6924 Sorengo, Switzerland; 7Clinica Santa Chiara, 6601 Locarno, Switzerland; 8S.H. Ho Urology Centre, Department of Surgery, Prince of Wales Hospital, The Chinese University of Hong Kong, Hong Kong, China; jeremyteoh@surgery.cuhk.edu.hk; 9Comprehensive Cancer Center, Department of Urology, Vienna General Hospital, Medical University of Vienna, Währinger Gürtel 18-20, 1090 Vienna, Austria; morikeiichiro29@gmail.com; 10Department of Urology, Jikei University School of Medicine, Tokyo 105-8461, Japan; 11Department of General, Minimally Invasive and Endocrine Surgery, Wroclaw Medical University, 50-556 Wroclaw, Poland; krzysztofkali@wp.pl

**Keywords:** upper tract urothelial carcinoma, tumor location, renal pelvis, ureter, radical nephroureterectomy, prognosis, oncological outcomes

## Abstract

Background: Upper tract urothelial carcinoma (UTUC) accounts for up to 10% of all urothelial neoplasms. Currently, various tumor-related factors are proposed to be of importance in UTUC prognostic models; however, the association of the primary UTUC location with oncological outcomes remains controversial. Thus, we sought to perform a systematic review and meta-analysis of the latest available evidence and assess the impact of primary tumor location on long-term oncological outcomes in patients with UTUC undergoing radical nephroureterectomy. Materials and Methods: A computerized systematic literature search was conducted in October 2021 through the PubMed, Web of Science, Scopus, and Cochrane Library databases. The primary endpoint was cancer-specific survival (CSS), and the secondary endpoints were overall survival (OS) and disease-free survival (DFS). Effect measures for the analyzed outcomes were reported hazard ratios (HRs) and 95% confidence intervals (CIs). Results: Among the total number of 16,836 UTUC in 17 included studies, 10,537 (62.6%) were renal pelvic tumors (RPTs), and 6299 (37.4%) were ureteral tumors (UTs). Pooled results indicated that patients with UT had significantly worse CSS (HR: 1.37, *p* < 0.001), OS (HR: 1.26, *p* = 0.003, and DFS (HR: 1.51, *p* < 0.001) compared to patients with RPT. Based on performed subgroup analyses, we identified different definitions of primary tumor location and geographical region as potential sources of heterogeneity. Conclusions: Ureteral location of UTUC is associated with significantly worse long-term oncological outcomes. Our results support the need for close follow-up and the consideration of perioperative chemotherapy in patients with UTUC located in the ureter. However, further prospective studies are needed to draw final conclusions.

## 1. Introduction

Upper tract urothelial carcinoma (UTUC) arising from the epithelium lining of the pyelocaliceal cavities or ureter accounts for 5–10% of all urothelial neoplasms [1]. Renal pelvic tumors (RPT) are approximately twice as common as ureteral tumors (UT) [2]. Regardless of the primary UTUC location, radical nephroureterectomy (RNU) with bladder cuff excision is considered the standard treatment of high-risk UTUC; however, the prognosis of patients with advanced disease is still unfavorable [2].

Novel prognostic factors are constantly being sought to improve treatment outcomes of patients with UTUC by allowing for more thorough planning of adjuvant treatment or follow-up strategies. To date, various tumor-related factors have been proposed by the European Association of Urology (EAU) to be of importance in UTUC risk stratification and prognostic models [2]. These include, e.g., tumor stage and grade, presence of lymphovascular invasion (LVI), or tumor architecture. The subject of tumor location has been also analyzed in the existing literature; however, the correlation of primary tumor location with oncological outcomes of UTUC remains controversial.

The previously conducted meta-analyses suggested that UT could be associated with worse prognosis compared to RPT [3,4]. In recent years, several new studies have contributed relevant information toward the prognostic implications of the primary UTUC location. Thus, we performed a systematic review and meta-analysis of the latest available evidence and assessed the impact of primary tumor location on long-term oncological outcomes in patients with UTUC undergoing RNU.

## 2. Materials and Methods

### 2.1. Search Strategy

The present systematic review and meta-analysis was performed according to the latest Preferred Reporting Items for Systematic Review and Meta-analysis (PRISMA) guidelines [5]. The protocol of this study was registered in priori with International Prospective Register of Systematic Reviews (PROSPERO), and it is available online (CRD42021245750).

Two investigators (Ł.N. and W.K.) independently performed a computerized systematic literature search in October 2021 through the PubMed, Web of Science, Scopus, and Cochrane Library databases. The Medical Subject Headings (MeSH) and non-MeSH terms were combined to create the following search string: (“upper tract urothelial carcinoma” OR ”upper tract urothelial cancer” OR “upper urinary tract cancer” OR “upper tract urothelial neoplasm” OR “transitional cell carcinoma of the upper urinary tract” OR “UTUC” OR “UUTC”) AND (“location” OR “locat*” OR “localization”) AND (“prognosis” OR “survival” OR “oncological outcomes”). The literature search was limited to English language articles without restrictions on publication year. Additional screening was performed from ahead of print articles published in the various urological journals. The references of the relevant review articles were also manually screened to ensure that no additional eligible studies were inadvertently omitted.

### 2.2. Inclusion and Exclusion Criteria

The population, investigated condition, comparison condition, outcome, and study design (PICOS) approach was used to define the eligibility criteria:(P)opulation: Studies that included patients with UTUC who underwent RNU.(I)nvestigated condition: Patients with tumors located in the ureter (UT group).(C)omparison condition: Patients with tumors located in the renal pelvis (RPT group).(O)utcome: The primary outcome was cancer-specific survival (CSS). The secondary outcomes were overall survival (OS) and disease-free survival (DFS). CSS was defined as the time from surgery to death from UTUC, while OS was defined as the time from surgery to death from any reason. DFS was defined as the time from the date of surgery to the date of documented relapse/recurrence at the surgery site, regional lymph nodes, and/or distant metastases.(S)tudy design: Randomized controlled trials (RCTs), nonrandomized observational cohorts, and population-based cohorts.

The studies were considered eligible if the primary UTUC location (RPT or UT) was defined as (1) tumor present only in the ureter or renal pelvis, or (2) tumor present only in the ureter or renal pelvis plus additionally dominant tumor within the ureter or renal pelvis in the case of multiple tumors. A dominant lesion was clarified as that with the highest pathological tumor stage.

Exclusion criteria were as follows: (1) noncomparative studies—reviews, letters, editorial comments, meeting abstracts, replies from authors, case reports; (2) studies including patients who underwent any type of kidney-sparing surgery; (3) studies on children, animal models, or cadaveric models; (4) studies reporting no sufficient data to estimate the hazard ratios (HRs) and 95% confidence intervals (CIs). For series published by the same authors or institutions, only the most recent or largest study was reported to reduce the risk of repeated data. Whenever two studies examined the same national database for overlapping periods of time, only the larger study was included.

### 2.3. Data Extraction

Two review authors (Ł.N. and W.K.) independently performed the data extraction process using a predefined template. Any disagreements or discrepancies were resolved by discussion with a senior coinvestigator who was not involved in the initial screening process (T.S.).

We extracted study-related data (first author, publication year, journal, country, study design, recruitment period, number of patients, and follow-up period), data on clinicopathological and treatment characteristics (age, gender, history of previous bladder cancer, performance of bladder cuff, pathological tumor stage and grade, proportion of patients with pathologically confirmed lymph node invasion (LNI), proportion of patients with concomitant carcinoma in situ (CIS), proportion of patients with positive LVI, proportion of patients who received perioperative systemic chemotherapy), and outcome measurements (HRs and 95% CIs associated with CSS, OS, DFS).

### 2.4. Quality Assessment and Risk of Bias

All selected nonrandomized studies were evaluated for their methodological quality using the Newcastle–Ottawa Scale (NOS). The “risk of bias” (RoB) for each included manuscript was assessed according to the principles outlined in the Cochrane Handbook for Systematic Reviews of Interventions [6]. The reports were initially evaluated in terms of allocation, sequence generation and concealment, blinding of participants, personnel and outcome assessors, completeness of outcome data, selective outcome reporting, and other sources of bias. In addition, the articles were reviewed based on their adjustment for the effects of the following confounders: age, pathological tumor stage, pathological tumor grade, concomitant CIS, LNI, and LVI. The risk of confounding bias was considered to be high if the confounder was not controlled for in multivariate analysis.

### 2.5. Statistical Analysis

All statistical analyses were performed using Review Manager 5.4 (The Nordic Cochrane Center, The Cochrane Collaboration, Copenhagen, Denmark), Statistica 13.3 (StatSoft Inc., Tulsa, OK, USA), and Stata 16.0 software (STATA Corporation, College Station, TX, USA).

Effect measures for the parameters of long-term survival (CSS, OS, DFS) were reported HRs and 95% CIs. For papers that did not directly provide HRs and 95% CIs, we extracted data from the presented Kaplan–Meier curves and calculated effect measures using methods described by Tierney et al. [7]. The statistical significance of the pooled HRs was evaluated by the Z test. Statistical pooling of the effect measures was based on the level of heterogeneity between the studies. Significant heterogeneity was indicated by either a ratio of >50% in I^2^ statistics or a *p*-value ≤ 0.10 in Cochran’s Q test, which led to the use of the random-effect model. Otherwise, the fixed-effect model was used in analysis. Publication bias for each comparison was assessed using Egger’s test. For all tests (other than Cochran’s Q test), a *p*-value ≤ 0.05 was considered a statistically significant difference.

## 3. Results

### 3.1. Literature Selection

The detailed flow diagram of study selection process (with subsequent exclusions) is presented in Figure 1. The literature search identified 854 references (224 from Pubmed, 469 from Scopus, 154 from Web of Science, and 7 from Cochrane Library). All citations were exported to the citation manager EndNote 20 (Clarivate Analytics), and duplicate references (*n* = 281) were removed. After screening of the titles and abstracts, 487 studies were excluded due to irrelevance to the present topic (*n* = 319) and study type (*n* = 168). The full texts of 38 studies were read in detail to determine their eligibility. In accordance with the study inclusion criteria, 17 and 6 manuscripts were excluded for insufficient outcome and inclusion of patients receiving kidney-sparing management, respectively. Eventually, there were 15 studies left, together with 2 studies selected from the references of relative articles. Thus, a total of 17 studies were included for final quantitative and qualitative synthesis [8,9,10,11,12,13,14,15,16,17,18,19,20,21,22,23,24].

### 3.2. Baseline Characteristics of Included Studies

The baseline characteristics of the included studies are presented in Table 1. All articles were published between 2009 and 2020. Selected papers comprised the total number of 16,836 UTUC, of which 10,537 (62.6%) were RPT, and 6299 (37.4%) were UT. All trials had retrospective design. Of the 17 studies, 9 [8,10,12,16,18,20,22,23,24] presented data from Asian populations, while the remaining articles reported data from European (3 studies [13,14,15]), North American (3 studies [9,11,19]), and international populations (2 studies [17,21]). The duration of follow-up (median or mean) ranged from 21 to 73 months. The quality scores of the selected manuscripts varied from 7 to 9 (indicating high methodological quality). Due to retrospective design, all included studies carried a high RoB. The issue of confounding through multivariate analyses was addressed in 13 [9,10,11,13,14,15,16,17,18,21,22,23,24] out of 17 reports (Figure 2).

### 3.3. Clinicopathological Characteristics of Included Studies

The clinicopathological characteristics of included articles are presented in Table 2. One study [10] provided clinicopathological data for the total cohort of patients (including both RPT and UT), and one study [19] did not report clinicopathological data for the analyzed subset of patients who underwent RNU. Male predominance was observed in the vast majority of papers. The average patient age in all reported cohorts was older than 60 years. In eight articles, RNU was performed in either the open or laparoscopic approach [10,12,17,18,20,21,22,24]. Ten out of 17 selected studies specified whether a bladder cuff excision was performed or not in particular study groups [8,9,10,11,13,14,17,18,21,22]. Of these 10 manuscripts, 9 reported a 100% rate of bladder cuff excision during RNU [8,9,10,13,14,17,18,21,22]. In several papers, a significantly higher proportion of ≥pT3 tumors was observed in the RPT group [11,13,16,17,18,20,21,23]. Only one study [24] reported a significantly higher proportion of ≥pT3 tumors in the UT group. The proportion of patients with positive LNI ranged from 3.1 to 9.8 and 2.3 to 12.7 in the RPT and UT groups, respectively [8,9,11,12,13,14,15,17,20,21,22,23,24], while two studies excluded patients with lymph node metastases [16,18]. The concomitant CIS rates were available in only six articles [8,9,10,20,21,22], of which three [9,20,22] reported significantly higher rates of concomitant CIS in UT groups. If reported, positive LVI rates were predominantly higher in the RPT group [12,13,15,21,23]. In almost all included studies, patients receiving systemic neoadjuvant chemotherapy (NAC) or adjuvant chemotherapy (AC) were initially excluded, or no data were available.

### 3.4. Meta-Analysis Results

Data for CSS were extractable from 14 studies [8,9,11,12,13,15,16,17,18,19,20,21,23,24]. Pooled results indicated that patients with UT had significantly worse CSS after RNU (HR: 1.37, 95% CI: 1.19–1.57, *p* < 0.001) compared to patients with RPT (Figure 3). Significant between-study heterogeneity was observed (*p* = 0.03 and I^2^ = 47%); thus, a random-effect model was used for data synthesis.

Data for OS were extractable from 7 studies [8,14,15,16,18,19,23]. A pooled results indicated that patients with UT had significantly worse OS after RNU (HR: 1.26, 95% CI: 1.08–1.46, *p* = 0.003) compared to patients with RPT (Figure 4). Significant between-study heterogeneity was observed (*p* = 0.08 and I^2^ = 47%), thus, a random-effect model was used for data synthesis.

Data for DFS were extractable from seven studies [12,15,17,18,21,22,23]. Pooled results indicated that patients with UT had significantly worse DFS after RNU (HR: 1.51, 95% CI: 1.20–1.89, *p* < 0.001) compared to patients with RPT (Figure 5). Significant between-study heterogeneity was observed (*p* = 0.09 and I^2^ = 46%); thus, a random-effect model was used for data synthesis.

The funnel plots were basically symmetrical for each outcome of interest (CSS, OS, and DFS) (Appendix A). In addition, results of Egger’s test did not demonstrate a significant publication bias (results are presented in Appendix A).

To explore the heterogeneity of the primary outcome (CSS), the prognostic value of primary UTUC location was evaluated by performing subgroup analyses stratified by type of Cox regression model, reported definition of primary UTUC location, and geographical location. As OS and DFS were reported in few cohorts (less than 10), no subgroup analyses were conducted for these oncological outcomes.

Stratification of studies by the type of used Cox regression model (univariable/multivariable) failed to explain between-study heterogeneity (Figure 6). On the contrary, we identified different definitions of primary tumor location (Figure 7) and geographical region (Figure 8) as potential sources of heterogeneity. Pooled effect measures of these subgroup analyses were consistent with primary findings (indicating worse CSS in UT group), except for populations of patients from North America. In this cohort of patients, insignificant difference between RPT and UT was observed in terms of CSS (HR: 1.10, 95% CI: 0.98–1.23, *p* = 0.12). Additionally, we performed subgroup analyses stratified by differences in concomitant CIS rates; combined effect measures were similar to the main analysis (Appendix A).

## 4. Discussion

In the present systematic review and meta-analysis, we demonstrated that ureteral location of UTUC was associated with significantly worse long-term oncological outcomes (CSS, OS, DFS) compared to RPT. Our results were based on synthesis of data originating from 17 studies including a total number of 16,836 patients [8,9,10,11,12,13,14,15,16,17,18,19,20,21,22,23,24].

Differentially worse outcomes in UT can be explained in several ways. The most convincing theories relate to the differences in the tumor surrounding environment. In the case of RPT, renal parenchyma, perirenal fat, and Gerota’s fascia may function as a natural barriers and main determinants of anatomical disease spread [25]. Moreover, compared to a relatively thin periureteral layer of muscular and fatty tissue, complete excision of adjacent tissues that possibly contain pathologically identifiable tumor cells or micro-metastases is more approachable in RPT. In some publications, the rate of positive surgical margins was reported to be significantly higher in patients with UT than in those with RPT, which could be directly associated with the higher risk of disease recurrence and worse CSS and OS [26]. Distinct disease characteristics between tumor location among patients with UTUC might also be related to differences in prognostic significance of LVI, as it is considered one of the crucial factors of cancer dissemination. Lee et al. showed that positive LVI represented the only significant predictor for CSS in patients with UT, but such a prognostic value of LVI was not observed in RPT (although the reported prevalence of LVI was higher in the latter location) [12]. Authors concluded that anatomical ureter features, such as thin wall containing complex network of blood vessels and lymphatic plexuses, enable easier spread of cancer cells [12]. In addition, certain genetic and epigenetic changes, such as increased promoter hypermethylation, may result in worse biological potential for ureteral tumors, as investigated in the recent literature [8]. Presence of concomitant CIS is another factor that potentially increases the aggressiveness of UTUC. As some of analyzed papers reported significantly higher rates of concomitant CIS in UT, it could be assumed that worse oncological outcomes in UT might result from adverse histopathological features rather than anatomical localization. However, we found that even if the concomitant CIS rates were similar between UTUC located in the renal pelvis or ureter, UTs were characterized by worse CSS. Because these results were based only on scarce data originating from a few papers, the influence of histopathological features (e.g., various concomitant CIS rates) on final results should not be fully excluded.

In several included studies, RNU was performed by either the open or laparoscopic method. The possible inferiority of laparoscopic RNU compared to open RNU in terms of oncological outcomes (especially in locally advanced UTUC) was postulated by some authors, as they hypothesized that manipulation within a tumor mass during elevated intra-abdominal pressure might increase the risk of disease recurrence [27]. Due to lack of data, we were unable to perform reliable subgroup analyses of particular UTUC locations stratified by the RNU approach. However, recent meta-analysis demonstrated that laparoscopic RNU and open RNU are comparable in terms of survival outcomes (CSS, OS, and recurrence-free survival (RFS)), even in patients with locally advanced UTUC [28].

It has to be emphasized that definition of primary tumor location differed among eligible studies. Some papers [8,9,11,12,13,16,17,18,19] included in our meta-analysis categorized multiple lesions in the upper urinary tract as primary RPT or UT based on the location of the dominant tumor (defined by highest pathological tumor stage, grade, or size), and other studies [15,20,21,23,24] analyzed only lesions present in particular locations. Because inappropriate grouping might mask the true impact of UTUC location on oncological outcomes, we conducted additional subgroup analysis stratified by two reported definitions. Similarly to the main analysis, we observed significantly worse long-term survival outcomes in the UT group regardless of the definition of the primary tumor location. However, we found significant reduction in between-study heterogeneity. It is currently postulated that multifocal tumors should be considered as a “third type” of UTUC location because of their poor prognosis compared to unifocal tumors [2,29]. Nevertheless, definitions of multifocal tumors are highly heterogeneous between existing studies (some of them requiring the involvement of both the renal pelvis and ureter, and others requiring only a multiple number of tumors); thus, we did not address this issue in detail.

Geographical diversity, including genetic, cultural, and environmental factors, was demonstrated to have significant influence on UTUC prevalence and characteristics [30]. Therefore, we hypothesized that observed heterogeneity between included studies may be associated with inclusion of various geographical populations. We showed that when compared to RPT, UT were characterized by significantly worse long-term survival outcomes in Asian and European populations (the results for European populations should be interpreted cautiously because of the limited number of patients). On the contrary, no significant differences between RPT and UT were found in cohorts of patients from North America. Potential explanation of these findings might be a higher exposure to aristolochic acid in Asian populations, as it was demonstrated to increase the risk of advanced UTUC [31].

The main findings of our meta-analysis are consistent with previously published reports. Initially, Wu et al. demonstrated that presence of UT was an independent prognostic factor of worse CSS and RFS in patients with UTUC treated with RNU [3]. Notably, in the analysis of the latter survival parameter, authors included papers with highly heterogeneous RFS definition, which lowers the reliability of the results [3]. Contrary to this study, we performed recurrence analysis based on strict and unequivocal criteria. Subsequently, Kaczmarek et al. showed that ureteral involvement of UTUC is significantly associated with worse CSS and OS [4]. However, despite the good methodological quality, this study lacked data regarding recurrence, as well as profound subgroup analyses explaining high heterogeneity of obtained results [4]. Therefore, for the first time, we provided a detailed analysis of long-term oncological outcomes based on several new stratification variables.

Based on the results of this systematic review and meta-analysis, more stringent follow-up schedules should be considered in cases of UT, as such tumors are at higher risk of disease recurrence compared to RPT. Currently, there are no strict criteria of including patients with UTUC for systemic treatment, which was demonstrated to have beneficial effects on survival outcomes [32]. Our results suggests that UTUC location should be considered as one of the potential clinical factors for administration of perioperative chemotherapy, as CSS and OS are significantly decreased. However, further studies analyzing the association between primary UTUC location in NAC and AC setting are required.

Our current study has several strengths. We included the latest available studies with representative populations and provided the most detailed exploratory subgroup analyses to date, which allowed us to explain substantial in-between study heterogeneity. However, our study also has notable limitations. First, the strength of the conclusions that can be drawn from our meta-analysis is still limited by the fact that all included studies were retrospective in nature, with their own inherent biases. Second, additional data regarding factors such as the RNU approach (open or laparoscopic), performance of bladder cuff excision, presence of concomitant CIS and LVI, and administration of adjuvant chemotherapy, were not uniformly reported, and the influence of such heterogeneity could not be fully neglected. Third, the adjustments for confounders in the Cox regression analyses were not uniform in the included trials, which might introduce additional bias. Fourth, the results of subgroup analyses should be interpreted carefully, as some of them were based on a limited number of patients.

## 5. Conclusions

Ureteral location of UTUC is associated with significantly worse long-term oncological outcomes compared to RPT. Our results support the need for close follow-up and the consideration of perioperative chemotherapy for patients with UT. However, further prospective studies are needed to draw the final conclusions.

## Figures and Tables

**Figure 1 jpm-11-01363-f001:**
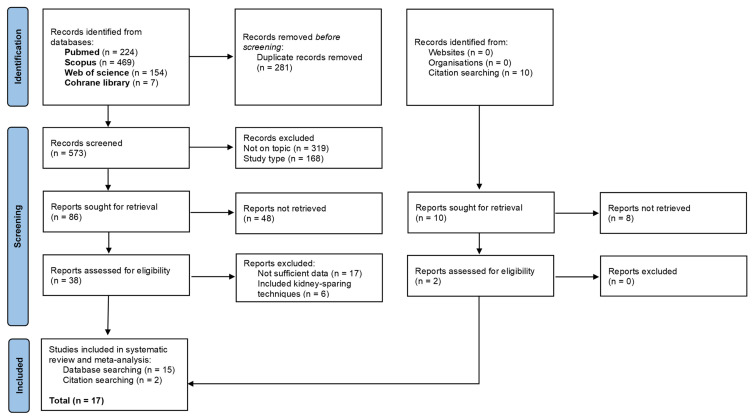
Preferred reporting items for systematic reviews and meta-analysis (PRISMA) flowchart.

**Figure 2 jpm-11-01363-f002:**
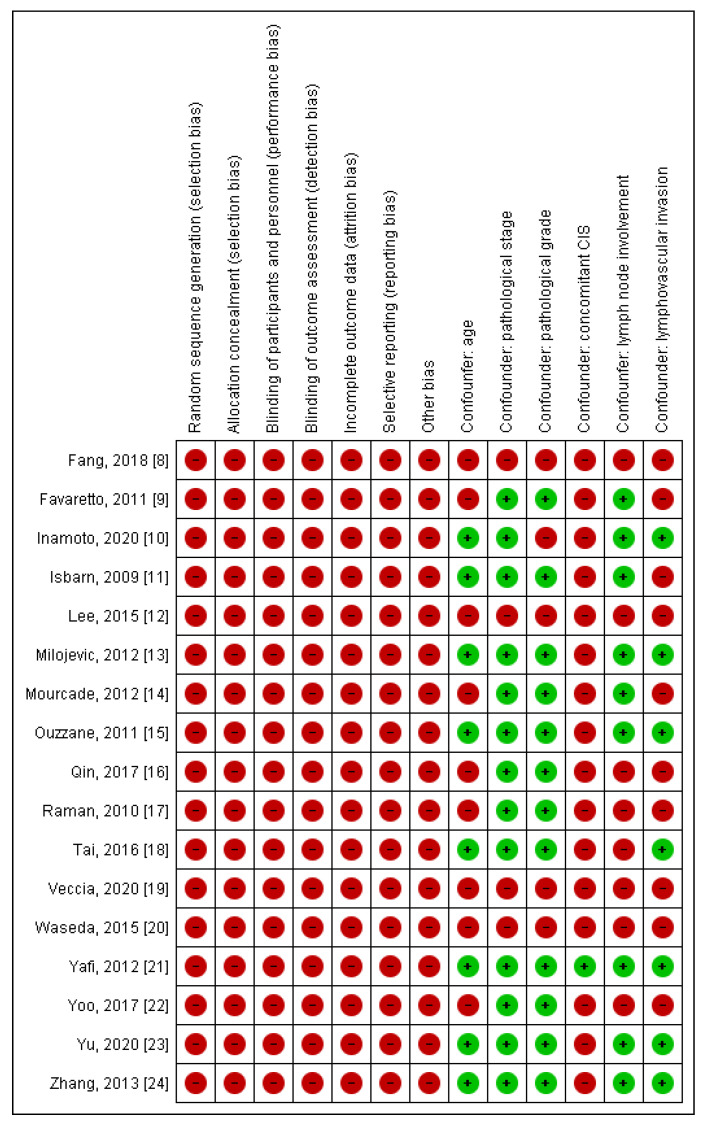
The risk of bias (RoB) and confounding assessment for all included studies [8,9,10,11,12,13,14,15,16,17,18,19,20,21,22,23,24]. CIS = carcinoma in situ.

**Figure 3 jpm-11-01363-f003:**
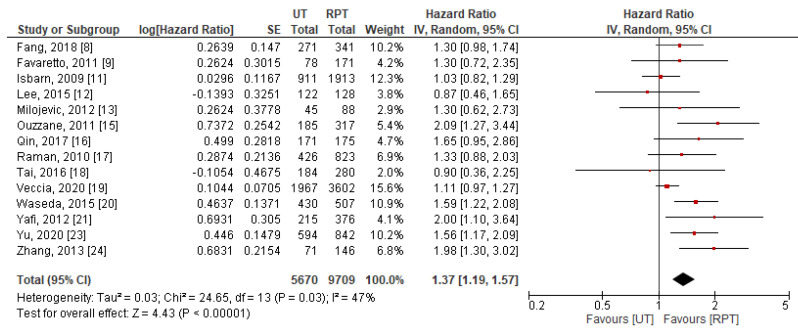
Forest plot and meta-analysis of cancer-specific survival [8,9,11,12,13,15,16,17,18,19,20,21,23,24]. CI = confidence interval; IV = inverse variance; RPT = renal pelvic tumor; SE = standard error; UT = ureteral tumor.

**Figure 4 jpm-11-01363-f004:**
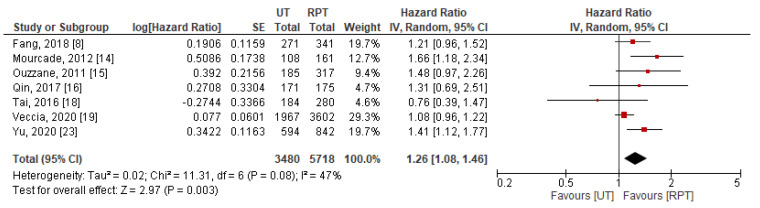
Forest plot and meta-analysis of overall survival [8,14,15,16,18,19,23]. CI = confidence interval; IV = inverse variance; RPT = renal pelvic tumor; SE = standard error; UT = ureteral tumor.

**Figure 5 jpm-11-01363-f005:**
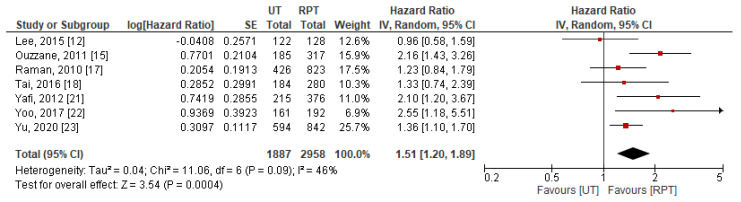
Forest plot and meta-analysis of disease-free survival [12,15,17,18,21,22,23]. CI = confidence interval; IV = inverse variance; RPT = renal pelvic tumor; SE = standard error; UT = ureteral tumor.

**Figure 6 jpm-11-01363-f006:**
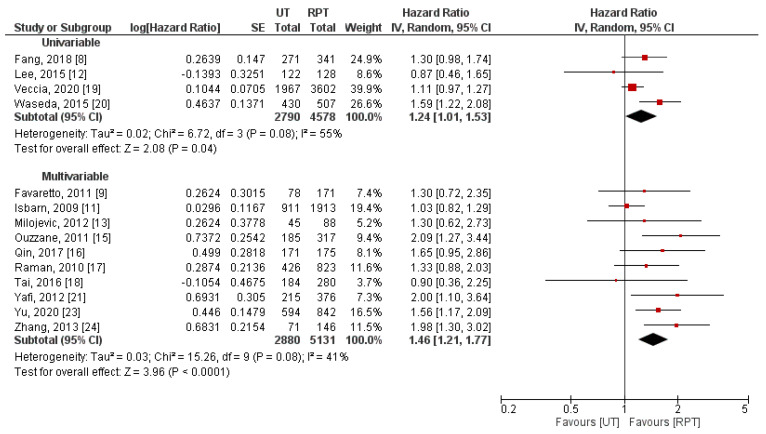
Forest plot and subgroup analysis of cancer-specific survival stratified by type of Cox regression model [8,9,11,12,13,15,16,17,18,19,20,21,23,24]. CI = confidence interval; IV = inverse variance; RPT = renal pelvic tumor; SE = standard error; UT = ureteral tumor.

**Figure 7 jpm-11-01363-f007:**
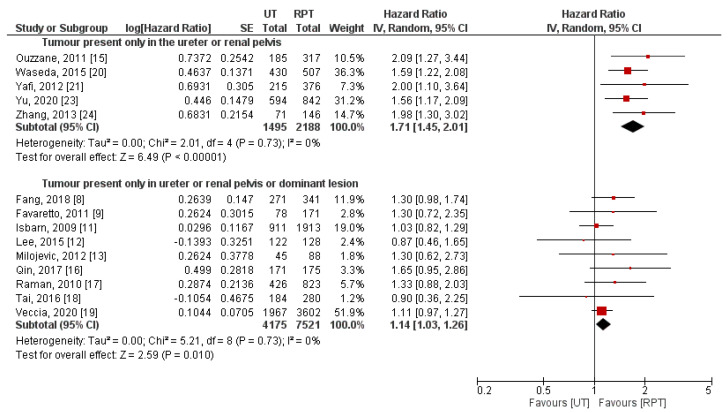
Forest plot and subgroup analysis of cancer-specific survival stratified by definition of primary UTUC location [8,9,11,12,13,15,16,17,18,19,20,21,23,24]. CI = confidence interval; IV = inverse variance; RPT = renal pelvic tumor; SE = standard error; UT = ureteral tumor.

**Figure 8 jpm-11-01363-f008:**
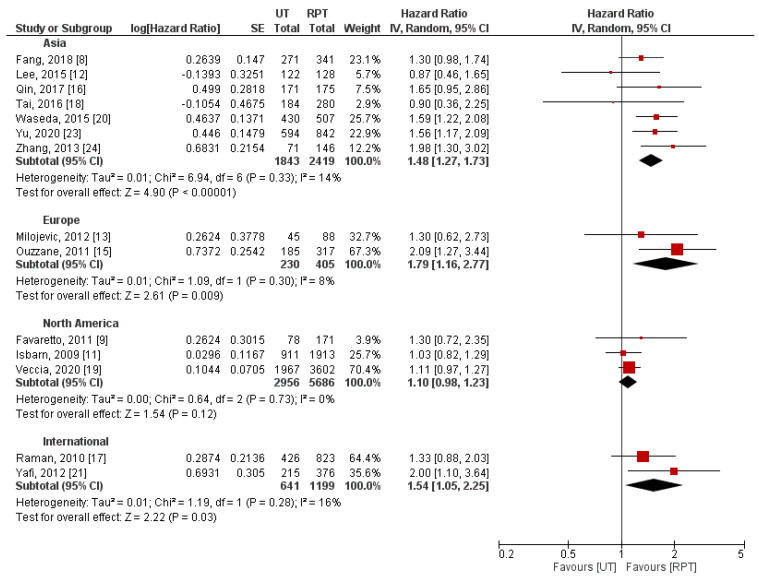
Forest plot and subgroup analysis of cancer-specific stratified by geographical location [8,9,11,12,13,15,16,17,18,19,20,21,23,24]. CI = confidence interval; IV = inverse variance; RPT = renal pelvic tumor; SE = standard error; UT = ureteral tumor.

**Table 1 jpm-11-01363-t001:** Baseline characteristics and quality assessment of included studies.

First Author, Year [Reference]	Journal	Country	Study Design	Recruitment Period, Years	No. of Patients, *n* RPT/UT	Follow Up, Months RPT/UT	Reported Outcomes	Methodological Quality (NOS)
Fang, 2018 [8]	*BMC Urology*	China	R, single-center	1999–2011	341/271	Median: 64	CSS, OS	8
Favaretto, 2011 [9]	*European Urology*	United States	R, single-center	1995–2008	171/78	Median: 48	CSS, DFS	8
Inamoto, 2020 [10]	*Current Urology*	Japan	R, multi-center	1994–2009	475/359 ^a^	Median: 34	DFS	7
Isbarn, 2009 [11]	*Journal of Urology*	United States	R, SEER database	1988–2004	1913/911	Median: 45/40	CSS	7
Lee, 2015 [12]	*Annals of Surgical Oncology*	Taiwan	R, single-center	2004–2010	128/122	Median: 41	CSS, DFS	8
Milojevic, 2012 [13]	*BJU International*	Serbia	R, single-center	1999–2009	88/45	Median: 35	CSS, DFS	7
Mouracade, 2012 [14]	*The Canadian Journal of Urology*	France	R, multi-center	1985–2005	161/108	Median: 70.3	OS	8
Ouzzane, 2011 [15]	*European Urology*	France	R, multi-center	1995–2010	317/185	Median: 32/30	CSS, OS, DFS	8
Qin, 2017 [16]	*Medicine (United States)*	China	R, single-center	2012–2016	175/171	Median: 21	CSS, OS	7
Raman, 2010 [17]	*European Urology*	Multinational	R, multi-center	1987–2007	823/426	Median: 49	CSS, DFS	8
Tai, 2016 [18]	*Urologic Oncology: Seminars and Original Investigations*	Taiwan	R, single-center	1996–2009	280/184	Median: 52	CSS, OS, DFS	8
Veccia, 2020 [19]	*International Journal of Urology*	United States	R, SEER database	2005–2015	3602 ^b^/1967 ^b^	Median: 29	CSS, OS	7
Waseda, 2015 [20]	*European Urology Focus*	Japan	R, multi-center	1995–2013	507/430	Median: 40	CSS	8
Yafi, 2012 [21]	*BJU International*	Multinational	R, multi-center	1990–2010	376/215	Median: 37/38	CSS, DFS	8
Yoo, 2017 [22]	*Clinical Genitourinary Cancer*	Korea	R, single-center	1998–2012	192/161	Mean: 73	DFS	8
Yu, 2020 [23]	*Journal of Clinical Medicine*	Taiwan	R, multi-center	1988–2019	842/594	Median: 33.6	CSS, OS, DFS	9
Zhang, 2013 [24]	*World Journal of Urology*	China	R, single-center	2000–2010	146/71	Median: 53/48	CSS, DFS	8

^a^ patients with ureteral tumors regardless of segmental location, ^b^ subset of patients who underwent radical nephroureterectomy. Abbreviations: CSS = cancer-specific survival; DFS = disease-free survival; NOS = Newcastle–Ottawa Scale; OS = overall survival; R = retrospective; RPT = renal pelvic tumor; UT = ureteral tumor.

**Table 2 jpm-11-01363-t002:** Clinicopathological characteristics of included studies.

First Author, Year [Reference]	Age RPT/UT	Male Gender, % RPT/UT	History of BC, % RPT/UT	Bladder Cuff Excision (%) RPT/UT	Pathological Stage ≥pT3, % RPT/UT	Pathological Grade (G3 or HG), % RPT/UT	LNI, % RPT/UT	Concomitant CIS, % RPT/UT	LVI, % RPT/UT	NAC, % RPT/UT	AC, % RPT/UT
Fang, 2018 [8]	65.3/68.1 ^a^ *	54.8/56.5	9.4/12.9	All patients	pT2–T4: 66.9/65.7	G3: 34.9/51.7 *	8.5/4.4	1.5/4.1	NR	Excluded	NA
Favaretto, 2011 [9]	71/73 ^b^	60.8/66.7	30.4/39.7	All patients	28.1/21.8	HG: 76.0/76.9	8.8/10.3	24.0/37.2 *	NR	Excluded	NA
Inamoto, 2020 [10]	72 ^a^	71.8	NR	All patients	32.7	G3: 47.0	4.3	13.1	27.7	Excluded	Excluded
Isbarn, 2009 [11]	71/72 ^b^ *	57.3/62.6 *	NR	62.5/83.0 *	57.9/38.3 *	NA	9.8/6.0 *	NR	NR	NR	NR
Lee, 2015 [12]	≥68 y: 50.0/61.5	42.2/44.3	Excluded	NR	37.5/29.5	HG: 76.6/79.5	7.8/6.6	NR	32.0/15.6 *	Excluded	16.8
Milojevic, 2012 [13]	66.7/66.6 ^a^	59.1/55.5	14.7/35.6 *	All patients	73.8/46.6 *	G3: 72.7/51.1 *	4.5/2.3	NR	70.5/35.6 *	Excluded	NA
Mouracade, 2012 [14]	67/66 ^a^	60.7/39.3	NR	All patients	31.7/36.1	G3: 31.0/28.7	8.1/8.3	NR	NR	NR	12.5/37.8 *
Ouzzane, 2011 [15]	69/71 ^b^	66.6/68.1	Excluded	NR	39.4/29.7	G3: 53.0/55.7	8.2/8.1	NR	20.5/15.1	Excluded	NR
Qin, 2017 [16]	≥68 y: 45.7/48.0	66.3/52.6 *	5.1/11.7 *	NR	pT2–T4: 29.1/21.6 *	HG: 82.9/83.0	Excluded	NR	NR	Excluded	54.3/43.3
Raman, 2010 [17]	68/69 ^b^	66.0/71.1	NR	All patients	38.2/21.8 *	HG: 58.9/62.4	6.7/4.9	NR	NR	NA	NA
Tai, 2016 [18]	67/69 ^b^	52.5/45.1	Excluded	All patients	37.5/20.2 *	HG: 50.7/64.1 *	Excluded	NR	17.5/17.4	Excluded	0/0
Veccia, 2020 [19]	NA	NA	NR	NR	NA	NA	NR	NR	NR	NR	NR
Waseda, 2015 [20]	69/70 ^b^	75.9/65.8	NR	NR	57.0/44.0 *	G3: 20.7/36.5 *	7.5/9.3	5.1/12.1 *	34.9/46.5 *	Excluded	NR
Yafi, 2012 [21]	68/69 ^b^	66.2/69.3	NR	All patients	31.1/20.5 *	G3: 54.5/54.9	3.7/2.3	10.1/12.6	18.1/10.7 *	Excluded	Excluded
Yoo, 2017 [22]	62.2/65.9 ^a^ *	76.6/70.2	NR	All patients	23.4/24.8	HG: 34.4/57.9 *	3.1/7.5	8.3/18.6 *	12.0/24.2 *	NR	Excluded
Yu, 2020 [23]	69.2/69.8 ^a^	43.0/40.6	5.3/5.1	NR	39.2/28.6 *	G3: 78.7/77.8	4.9/3.3	NR	22.4/15.3 *	NR	NR
Zhang, 2013 [24]	63/72 ^b^	58.2/63.4	NR	NR	43.2/60.6 *	G3: 70.5/49.3	6.8/12.7	NR	35.6/67.6 *	Excluded	NR

^a^ mean, ^b^ median, * Statistically significant difference between RPT and UT groups. Abbreviations: AC = adjuvant chemotherapy; BC = bladder cancer; CIS = carcinoma in situ; HG = high grade; LG = low grade; LNI = lymph node invasion; NA = not applicable; NAC = neoadjuvant chemotherapy; NR = not reported; RPT = renal pelvic tumor; UT = ureteral tumor.

## Data Availability

Not applicable.

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
