# Peer review of "The Impact of Primary Tumor Location on Long-Term Oncological Outcomes in Patients with Upper Tract Urothelial Carcinoma Treated with Radical Nephroureterectomy: A Systematic Review and Meta-Analysis"

_jpm, 2021, doi:10.3390/jpm11121363_

Round 1
Reviewer 1 Report
The authors of this article aim to assess the impact of the location of a urothelial tumor upon patient prognosis after radical nephroureterectomy. The article is well written, the methodology is well described and the results are clearly presented. I congratulate the authors for their work.
I think it would merit some discussion the type of surgical approach for RNU - as there is a potential risk for tumor seeding in patients with locally advanced tumors who undergo minimally-invasive surgery. Could this be potential factor of bias?
Author Response
Dear Editor,
in reference to the decision of minor revisions for the jpm-1454990 manuscript, we are submitting a revised version of the article. All issues raised by the Reviewers have been meticulously corrected. A detailed report on the amendments is presented below.
Reviewer 1 (yellow highlights)
- “I think it would merit some discussion the type of surgical approach for RNU - as there is a potential risk for tumor seeding in patients with locally advanced tumors who undergo minimally-invasive surgery. Could this be potential factor of bias?”
Our response: We highly appreciate Reviewers’ feedback and valuable comment. Indeed, the possible inferiority of laparoscopic RNU compared to open RNU in terms of oncological outcomes (especially in locally advanced UTUC) was postulated by some authors. Due to lack of data, we were unable to perform reliable subgroup analyses of particular UTUC locations stratified by RNU approach. However, our recent meta-analysis demonstrated that laparoscopic RNU and open RNU are comparable in terms of survival outcomes even in patients with locally advanced UTUC. We raised above mentioned issues in discussion section (Line: 132-140) and limitations section (Line: 191 - 195).
Sincerely,
Authors
Reviewer 2 Report
In the metanalysis and systematic review conducted by Krajewski et al, Authors aimed to assess the association between primary neoplasia location and long-term oncological outcomes in patients who are affected by UTUC and underwent radical nephroureterectomy.
Even though several factors were suggested by EAU as tumour-related in risk stratification models, the relationship remains unclear.
By the analysis of current literature and Forest plot construction, Authors finally stated that ureteral location of UTUC was concretely associated with substantially worse long-term oncological outcomes (Overall Survival, Cancer Free Survival, Disease Free Survival) compared to RPT. Moreover, Authors took care to eliminate any potential publication bias by application of Egger’s test
- Authors reported that Cis rates were available in only 6 out of 17 final articles; only 50% of them (3 studies) registered significantly higher rates of Cis in Ureteral Tumours. We think these are quite substantial data that deserve to be deepened especially to evaluate if the reported higher aggressiveness of UT could be resulting from a specifical histopathology rather than anatomical localization.
In conclusion the article has several strong points as the examination of the most recent evidence and the sub-stratification of the cohort analyzed. However, it is based on retrospective studies, whose results are easily distorted by bias.
Please pay attention to references, in particular number 25 that has been changed with 26.
Author Response
Dear Editor,
in reference to the decision of minor revisions for the jpm-1454990 manuscript, we are submitting a revised version of the article. All issues raised by the Reviewers have been meticulously corrected. A detailed report on the amendments is presented below:
Reviewer 2 (green highlights)
1. “Authors reported that Cis rates were available in only 6 out of 17 final articles; only 50% of them (3 studies) registered significantly higher rates of Cis in Ureteral Tumours. We think these are quite substantial data that deserve to be deepened especially to evaluate if the reported higher aggressiveness of UT could be resulting from a specifical histopathology rather than anatomical localization”
Our response: We highly appreciate Reviewers’ feedback and valuable comments. We totally agree that observed differences in histopathological features (e.g. different concomitant CIS rates) between RPT an UT groups can alter the results. Thus, we performed another subgroup analysis stratified by reported concomitant CIS rates (Line 88 - 90; Supplementary Figure 2). However, even if the reported CIS rates were similar between both groups, UT was characterized by worse CSS. We raised this issue in discussion section (Line 123 - 131) and limitations section (Line 191 - 194).
2. “Please pay attention to references, in particular number 25 that has been changed with 26”
Our response: References were checked again and they are correct.
Sincerely,
Authors